# A versatile automated pipeline for quantifying virus infectivity by label-free light microscopy and artificial intelligence

Anthony Petkidis [1,2,4], Vardan Andriasyan [1,4], Luca Murer [1,3], Romain Volle [1] & Urs F. Greber [1] ✉

Virus infectivity is traditionally determined by endpoint titration in cell cultures, and requires complex processing steps and human annotation. Here we developed an artificial intelligence (AI)-powered automated framework for ready detection of virus-induced cytopathic effect (DVICE). DVICE uses the convolutional neural network EfficientNet-B0 and transmitted light microscopy images of infected cell cultures, including coronavirus, influenza virus, rhinovirus, herpes simplex virus, vaccinia virus, and adenovirus. DVICE robustly measures virus-induced cytopathic effects (CPE), as shown by class activation mapping. Leave-one-out cross-validation in different cell types demonstrates high accuracy for different viruses, including SARS-CoV-2 in human saliva. Strikingly, DVICE exhibits virus class specificity, as shown with adenovirus, herpesvirus, rhinovirus, vaccinia virus, and SARS-CoV-2. In sum, DVICE provides unbiased infectivity scores of infectious agents causing CPE, and can be adapted to laboratory diagnostics, drug screening, serum neutralization or clinical samples.

Viruses affect cells in many different ways, including metabolism, signal transduction, gene expression, intracellular membrane organization, cytoskeletal integrity, and overall morphology[1–3]. Collectively, these changes are known as the cytopathic effect (CPE). CPE can be highly pathogen-specific, bearing diagnostic potential[4,5]. CPE is a hallmark of acute virus infection, and its detection is key for biological titer determination of inocula, as exemplified by plaque assay or endpoint dilution assays yielding tissue culture infectious dose 50 ($TCID_{50}$) values[6–9]. In the laboratory, these assays have traditionally been performed using cell stains, for example crystal violet (CV), or nuclear dyes, such as the DNA-intercalating Hoechst compounds[10–12]. In clinical settings, $TCID_{50}$ assays are not routinely used due to the requirement of manual annotation, the lack of virus specificity, and a rather slow readout that can take several days. Here we deliver a robust procedure to massively improve accuracy, automation, and marker-free infection detection. The procedure is based on light microscopy and AI and delivers virus-type-specific results. Light microscopy is suitable to study infected cells in live mode. It monitors changes in shape, morphology, and physiological state of individual cells or population of cells, and is suitable to assess infection variability[13–16]. In the past decade, automatic interpretation of microscopy images has been increasingly enhanced by deep learning (DL) and convolutional neural networks (CNNs) and enabled numerous applications in cell and infection biology[17,18]. For instance, transmitted light microscopy combined with DL predicts fluorescent labels[19,20], or classifies cell state and type[21]. Recent efforts have combined label-free imaging methods with image processing and artificial intelligence (AI) for automated detection of viral CPE in populations of cultured cells. For example, Hochdorfer and colleagues described that an automated image processing pipeline assessing cell confluency in transmitted light images

[1]Department of Molecular Life Sciences, University of Zürich, Winterthurerstrasse 190, 8057 Zürich, Switzerland. [2]Life Science Zurich Graduate School, ETH and University of Zürich, 8057 Zurich, Switzerland. [3]Present address: Roche Diagnostics, Forrenstrasse 2, 6343 Rotkreuz, Switzerland. [4]These authors contributed equally: Anthony Petkidis, Vardan Andriasyan. ✉e-mail: urs.greber@mls.uzh.ch

can be used to detect infection of BHK-21 cells with vesicular stomatitis virus (VSV)[22]. Unfortunately, this investigation was limited to only one virus and one cell line. Similarly, work by Wang and colleagues showed that DL can be employed for the detection of influenza-induced CPE in MDCK cells[23]. CPE detection by neural networks has also been described for influenza virus, parainfluenza virus, and enterovirus[24], but experimental conditions were not documented, and the code or dataset is not available, limiting broader useability. Another study proposed that DL can be used for early detection of viral CPE[25], but this approach requires a specifically trained model for a given cell line, virus, and imaging modality, making it difficult to use. Here we present a broad framework for detection of virus-induced cytopathic effect (DVICE) to score CPE in populations of cultured cells under well-defined experimental settings. We employ the recent CNN EfficientNet-B0[26] to achieve robust infection detection for a panel of different cell lines and viruses. Our procedure is compatible with live-cell imaging. It specifically recognizes image regions associated with CPE and opens new ways to standardization and automation of virus infectivity measurements.

## Results

### Automated transmitted light microscopy combined with AI-based image classification enables high-throughput virus infection readout

To establish a workflow for automated scoring of virus infections, we performed a serial dilution infection protocol, and annotated infection phenotypes by transmission light (TL) microscopy and crystal violet (CV) staining to obtain ground truth data. The training of DVICE was done in the next step, and the results were compared afterward (Fig. 1A). Ten thousand permissive cancer cells were seeded into 96-well plates, followed by inoculation with either human adenovirus species C type 5 (short AdV), herpes simplex virus type 1 (short HSV), influenza A virus (IAV), rhinovirus type A16 (short RV), vaccinia virus-WR (short VACV), coronavirus (CoV)−229E, CoV-OC43, or several isolates of severe acute respiratory syndrome CoV-2 (short SARS-CoV-2), and incubated cells for 7 days to allow for manifestation of CPE. TL images were acquired using a high-throughput microscope ImageXpress Micro Confocal (IXM-C, Molecular Devices) with a ×4 magnification objective and a plate loading robot. One central site was imaged for each well, covering approximately one-third of the well. Cells were

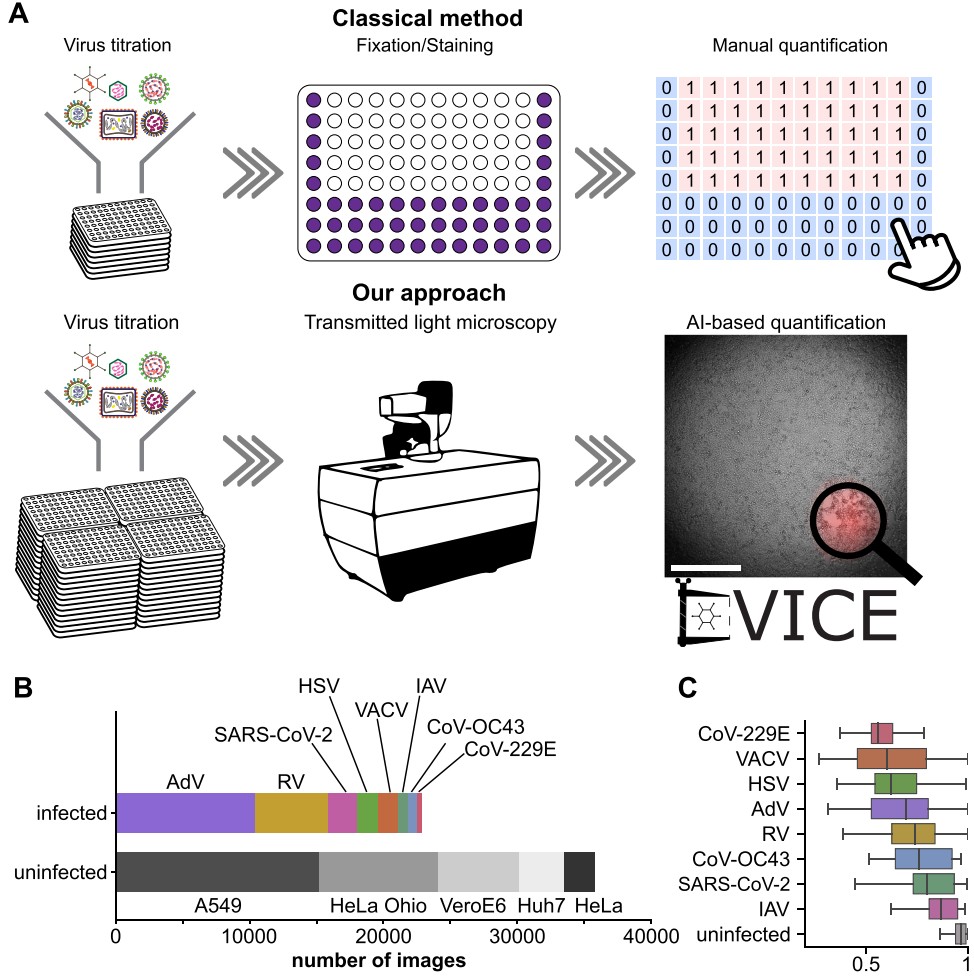

**Fig. 1 | Workflow for automated readout of viral infection and dataset composition. A** Classical method (top) for infection readout employs crystal violet staining followed by manual annotation of virally induced lesions in a cell monolayer. Our proposed approach (bottom) uses automated image acquisition and AI-based detection of virus-induced cytopathic effect (DVICE). The red overlay indicates areas of network attention. Scale bar 1 mm. **B** Composition of acquired dataset, indicating the proportions of viruses in the images of infected wells, and cell lines for uninfected wells. **C** Quantification of cell confluency for different viruses and for uninfected images. Lines show the medians of the distributions, boxes show the quartiles, and whiskers are drawn to the farthest datapoint within 1.5*inter-quartile range (IQR) from the nearest hinge. CoV-229E: $n = 342$, VACV: $n = 1520$, HSV: $n = 1561$, AdV: $n = 10,422$, RV: $n = 5466$, CoV-OC43: $n = 675$, SARS-CoV-2: $n = 2160$, IAV: $n = 722$, uninfected: $n = 35,743$. Source data are provided as a Source Data file.

then fixed with 4% paraformaldehyde (PFA) for 30 min, and stained with 0.25% CV. The infection of the stained samples was independently annotated by three human experts. Infection-annotated images were then used to train a CNN based on the EfficientNet-B0 architecture[26] for infection state classification. The number of images used from the different viruses comprised a total of 58,619 images, of which 22,873 images (39%) were from infected, and 35,746 images (61%) from uninfected wells (Fig. 1B). The dataset comprised five different cell lines, human lung epithelial A549 cells for infection with AdV, HSV, IAV, VACV, human cervical cancer HeLa-ATCC cells for VACV, HeLa-Ohio cells for RV, human hepatoma Huh7 cells for CoV-229E and CoV-OC43, and African green monkey VeroE6 cells for SARS-CoV-2. Example images are provided in Supplementary Fig. 1, and viruses, cell lines, and other reagents are listed in Table 1. To enhance the infection readout of SARS-CoV-2, we used three transgenic cell lines expressing the angiotensin-converting enzyme 2 (ACE2) or the transmembrane protease (TMPRSS2), namely A549-ACE2[27], Huh7-ACE2[27], and VeroE6-TMPRSS2[28]. We also acquired a total of 3840 images from uninfected sparsely seeded A549, HeLa-ATCC, HeLa-Ohio, and Huh7 cells, 960 images from each cell line.

As previous work demonstrated that cell confluency can be a surrogate for the infection state[22], we next quantified the cell confluency in our dataset (Fig. 1C, Supplementary Fig. 2). As expected, the median confluency was highest for uninfected wells. Yet, it showed high inter- and intra-class variability for the different conditions. Intraclass variability may arise from the range of viral concentrations used in the serial dilution assays, and interclass variability can be due to the virus-specific manifestation of CPE.

## A convolutional neural network achieves human-level accuracy in infection readout and identifies infection-specific features

We used human expert annotation of the CV-stained plates as ground truth for the infection state and thereupon trained a light-weight CNN based on the EfficientNet-B0[26] architecture to classify the TL images of infected and uninfected wells. For network training, images were downscaled to a size of 224 × 224 pixels using bicubic interpolation. We trained DVICE for the classification of infected images and compared its performance to several conventional machine learning (ML) algorithms, including support vector machine (SVM), $k$-nearest neighbors ($k$-NN), Gaussian naive Bayes (GNB), decision tree (DT) classifier, logistic regression (LR), and random forest (RF) classifier. DVICE achieved an area under the receiver operating characteristic curve (AUROC) of $0.991 \pm 0.001$, surpassing canonical ML methods (Fig. 2A, Supplementary Fig. 2A). Conventional ML methods were trained on histograms of oriented gradients (HOG)[29], which demonstrated superior performance compared to confluency- or intensity-based input features (Supplementary Figs. 3 and 4).

Next, we assessed the suitability of DVICE for application in the readout of $TCID_{50}$ plates. Entire virus titration plates were withheld from the training and validation set, and the trained model was used for infection state classification and subsequent $TCID_{50}$ calculation using the specific infection (SIN) method[30]. Compared with human annotation, DVICE achieved a squared Pearson correlation coefficient of $R^2 = 0.986$ (slope $1.00 \pm 0.01$, $n = 130$, $p = 10^{-120}$), indicating excellent agreement between actual and predicted labels (Fig. 2B, C). To determine whether DVICE learned robust features for infection scoring, we used a procedure known as class activation mapping (CAM)[31] to visualize important regions for infection detection (Fig. 2D). This procedure harnesses the global average pooling (GAP) layer of the EfficientNet-B0 architecture, which yields a spatial feature map. This feature map can be upscaled and overlaid with the originally acquired image (Fig. 2D). In images with confined regions of CPE, the network attention was typically focused on regions with virus-induced lesions. Interestingly, localization was still preserved despite the heavy image resizing to ~1% of the original pixel count. The network had a tendency

towards disregarding the dark image corners and had a flat attention map for uninfected wells. These CAM analyses show that DVICE recognizes robust features associated with CPE, and thereby enables reliable infection detection.

## DVICE is suitable for real-time infection monitoring and transferable to different imaging modalities

As our framework does not require chemical fixation of the sample and is compatible with live-cell imaging, we hypothesized that DVICE can be used for real-time monitoring of virus infections. To test this, ten thousand A549 cells were seeded in each well of a 96-well plate overnight and infected with AdV-IX-FS2A-GFP, which expresses GFP under the control of the promoter of the intermediate-late viral protein IX[32]. TL and GFP fluorescence images were acquired each day until day 7 post-infection (pi) (Fig. 3A). Virus concentration affected both the onset and magnitude of GFP expression. The onset of GFP expression generally preceded CPE detected by DVICE, suggesting that DVICE scores features late in infection but not early ones when cells are still fully attached and do not show lesions in the CV staining. Importantly, DVICE did not score sparse cells as infected, despite a correlation between infection state and confluency in the training dataset (see Fig. 1C). We attribute this to the presence of images of sparsely seeded cells in our dataset, enabling the network to learn that low cell confluency is not a defining hallmark of viral infection state. In summary, DVICE can be used to monitor the progression of infection in live cells and without interference.

A frequent limitation in applications of neural networks is a lack of generalization beyond the conditions of training[33,34]. To address this issue, we performed leave-one-out cross-validations, where we trained and validated the network on all images, including uninfected samples, while withholding images from a given virus. The performance of the network was then assessed on the withheld images. Results showed high AUROC values > 0.7 up to near 1 (Fig. 3B), suggesting that the performance variability for the different viruses could be attributed to the nature of the CPE or the number of images in the particular training dataset. Notably, however, the overall high AUROC values in the leave-one-out cross-validation indicate good generalization, which favors the DVICE application to new settings. Accordingly, the CV staining for HSV showed a better spatial separation between infected and uninfected wells compared to AdV (Supplementary Fig. 1C and D). These results suggest that a fast replicating virus, such as HSV[35], gives rise to distinct CPE compared to a somewhat slower replicating virus, such as AdV species C, for example C2 or C5[36].

To further assess the versatility of DVICE, TL images from infected cells were acquired with two different microscopes, the IXM-C (Molecular Devices), which was used to record the training dataset, and the Cytation 5 microscope (Agilent). Ten thousand A549 cells were seeded per well in 96-well plates and infected with serial dilutions of AdV, HSV, or VACV or were left uninfected. At 7 dpi, DVICE achieved an AUROC of $0.873 \pm 0.071$ for images acquired at the Cytation 5 microscope, compared to $0.941 \pm 0.004$ for the IXM-C (Fig. 3C). The predictions of DVICE for images acquired at the IXM-C and the Cytation 5 were in excellent agreement, as reflected in a value of 0.92 for Krippendorff's alpha[37].

## High accuracy of DVICE at scoring infectious SARS-CoV-2 in human samples

As DVICE successfully scored viral infectivity, we tested the possibility that our workflow could detect viral infectivity in clinical samples. We spiked samples of human saliva from a PCR-negative donor with a laboratory SARS-CoV-2 BA.1 variant stock reaching a virus titer corresponding to genome equivalents seen in hospitalized COVID-19 patients[38,39]. Samples were diluted with DMEM, passed through a 0.22 μm filter to remove cellular debris as well as bacteria, and then incubated at different temperatures for different periods of time, followed by biological titer determination in $TCID_{50}$ assays (Fig. 4A). The

**Table 1 | Key reagents and resources used in this study**

| Reagent or resource | Source | Identifier |
|---|---|---|
| **Bacterial and virus strains** | | |
| Adenovirus C5 | Kindly provided by Silvio Hemmi (University of Zurich, Switzerland) | |
| Adenovirus C5-IX-FS2A-GFP | Kindly provided by Silvio Hemmi (University of Zurich, Switzerland)[45] | https://doi.org/10.1016/j.isci.2021.102543 |
| hCoV-229E-GFP | Kindly provided by Dr. Volker Thiel (University of Bern, Switzerland)[64] | https://doi.org/10.1128/mBio.00171-10 |
| hCoV-OC43 | American Type Culture Collection (ATCC) | Cat #VR-1558 |
| SARS-CoV-2 BA.1 (B.1.1.529.1) | Obtained from the RIVM (Netherlands) through European Virus Archive global[43] | NH-RIVM-71076/2021 |
| RV-A16 | Luca Murer[51] | N/A |
| VACV_WR E/L-GFP | Kindly provided by Jason Mercer (University of Birmingham, UK) | |
| HSV-1-C12-CMV-GFP | Kindly provided by Stacey Efstathiou (University of Cambridge, UK)[35,55,56] | |
| Influenza A virus (IAV) H1N1 WSN | Kindly provided by Yohei Yamauchi[35] | N/A |
| **Cell culture reagents** | | |
| DMEM medium | Sigma-Aldrich | Cat #D6429 |
| Non-essential amino acids (NEAA) | Sigma-Aldrich | Cat #M7145 |
| Fetal bovine serum (FBS) | Gibco | Cat #10270-106 |
| Penicillin–streptomycin | Sigma-Aldrich | Cat #P0781 |
| Trypsin-EDTA | Sigma-Aldrich | Cat #C-41020 |
| PBS buffer w/o $Ca^{2+}$ and $Mg^{2+}$ | Animated/Bioconcept | Cat #3-05P29-M |
| Blasticidin | InvivoGen | Cat #ant-bl-1 |
| Geneticin | Merck | Cat #G418-RO |
| Chemicals, peptides, and recombinant proteins | | |
| TRIzol Reagent | Invitrogen | Cat #15596026 |
| **Molecular biology, RT-qPCR** | | |
| Direct-zol RNA Miniprep kit | Zymo Research | Cat #R2050 |
| Deposited data | | |
| Dataset | This paper | |
| DVICE models | This paper | |
| **Experimental models: Cell Lines** | | |
| Monkey: VeroE6 | Kindly provided by Dr. Volker Thiel (University of Bern, Switzerland) | |
| Monkey: VeroE6-TMPRSS2 | Kindly provided by Dr. Volker Thiel (University of Bern, Switzerland) | NIBSC 100978 |
| Human: Huh7 | Kindly provided by Dr. Volker Thiel (University of Bern, Switzerland) | |
| Human: Huh7-ACE2 | Laboratory-made by stable transfection with a lentivector (pLVX-ACE2-IRES-BSD)[27] | N/A |
| Human: HeLa | American Type Culture Collection (ATCC) | |
| Human: HeLa Ohio | Obtained from Laurent Kaiser, Central Laboratory of Virology, University Hospital Geneva, Switzerland | ECACC 84121901 |
| Human: A549 | American Type Culture Collection (ATCC) | ATCC CCL-185 |
| Human: A549-ACE2 | Laboratory-made by stable transfection with a lentivector (pLVX-ACE2-IRES-BSD)[27] | N/A |
| **Software and algorithms** | | |
| Anaconda Python v3.9.7 | Anaconda, Inc. | https://www.anaconda.com/ |
| Tensorflow v2.7.0 | Abadi et al. [57] | https://github.com/tensorflow/tensorflow |
| scikit-learn v1.2.2 | Pedregosa et al. [61] | https://github.com/scikit-learn/scikit-learn |
| Ilastik v1.4.0 | Berg et al. [62] | https://www.ilastik.org/download.html |
| DVICE | This paper | |
| **Other** | | |
| GeForce RTX 3090 | Nvidia | N/A |
| Automated high-throughput microscope ImageXpress Confocal Micro (IXM-C) | Molecular Devices | N/A |
| Automated imaging microplate reader BioTek Cytation 5 | Agilent | N/A |

presence of saliva reduced the viral titer in a time- and temperature-dependent manner compared to virus in DMEM only, but infectious titer was readily detectable in all conditions, providing proof-of-concept for investigation of clinical specimens. Human annotation of infection was compared to DVICE for a total of 3646 images (including 627 infected images) of A549-ACE2, Huh7-ACE2, or VeroE6-TMPRSS2 cells inoculated with serial dilutions of saliva spiked with SARS-CoV-2 BA.1, where DVICE achieved an AUROC of 0.918 ± 0.020. DVICE showed

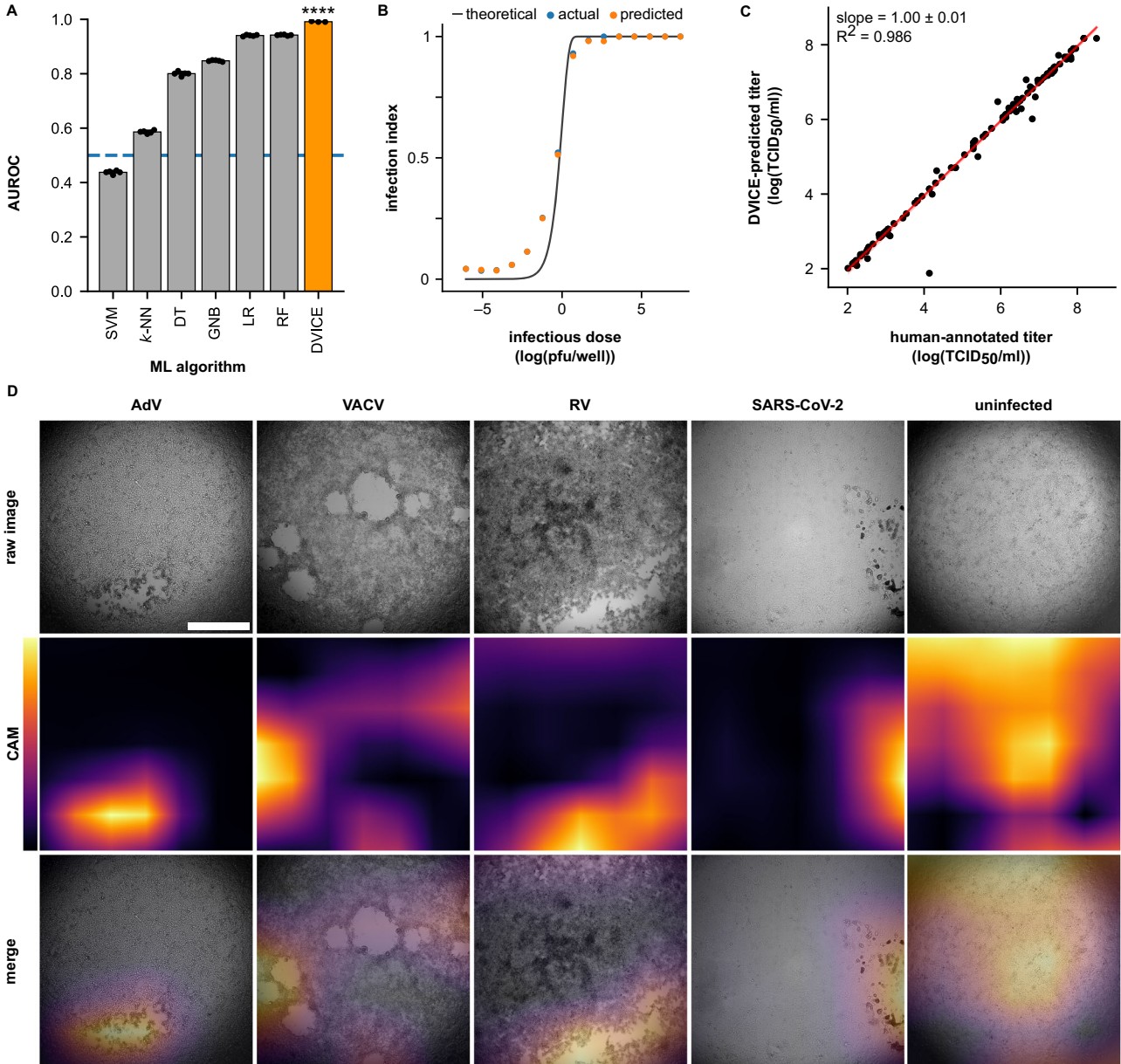

**Fig. 2 | Network performance and characteristics. A** Area under the receiver operating characteristic curve (AUROC) for different machine learning algorithms, including support-vector machine (SVM), *k*-nearest neighbors (*k*-NN), Gaussian naive Bayes (GNB), decision tree (DT), logistic regression (LR), random forest (RF), and DVICE. A nonparametric Kruskal–Wallis test with Dunn's correction for multiple testing was performed to evaluate the conventional ML algorithms against DVICE. SVM, *k*-NN, GNB, DT, LR, RF: $n = 5$, DVICE: $n = 3$. Data are presented as means and error bars indicate standard deviations. Adjusted *p*-value: ****$p = 0.0000276414$. **B** Dependency of infection index on virus concentration. The concentration of plaque-forming units (pfu) was obtained from the plate annotation and well position. Pfu values were grouped into 15 bins, and the plotted points indicate the bins' mean values. Actual values were annotated by human experts, and predicted values were provided by DVICE. log is the logarithm base 10. The theoretical curve is provided by the Poisson distribution. $n = 12,640$. **C** Comparison of humanly annotated (actual) and predicted TCID$_{50}$ values with linear regression line (red). The shaded region shows the 99.9% confidence interval of the regression curve. DVICE achieved a squared Pearson correlation coefficient of $R^2 = 0.986$ (slope $1.00 \pm 0.01$, $p = 10^{-120}$). $n = 130$. **D** Example images of different viruses and class activation maps (CAMs), indicating regions of network attention for recognition of virus infection. Scale bar 1 mm. Source data are provided as a Source Data file.

a sensitivity (true positive rate) of $0.946 \pm 0.006$ and specificity (true negative rate) of $0.890 \pm 0.044$ (Fig. 4B). These data suggest that DVICE provides reliable results with human samples, and can be potentially considered for virus titer determination in a clinical context, for example, antiviral drug efficacy studies.

**DVICE distinguishes infections by different viruses**
As DVICE recognizes viral CPE, we next tested whether it could also be extended to detect the particular nature of the infecting virus. As not

all cell lines in our study are susceptible to infection with all viruses, the cell line information could provide cues about the infecting virus. To mitigate any cell line-specific information and incentivize the network to learn the virus-specific infection signature, we trained DVICE on the previously generated segmentation maps of the images. Example images of the segmentation maps are provided in Fig. 5A. DVICE was trained with the same procedure as described above, using a dense layer with six output classes (uninfected, AdV, HSV, RV, SARS-CoV-2, VACV) and categorical instead of binary cross entropy loss. Overall,

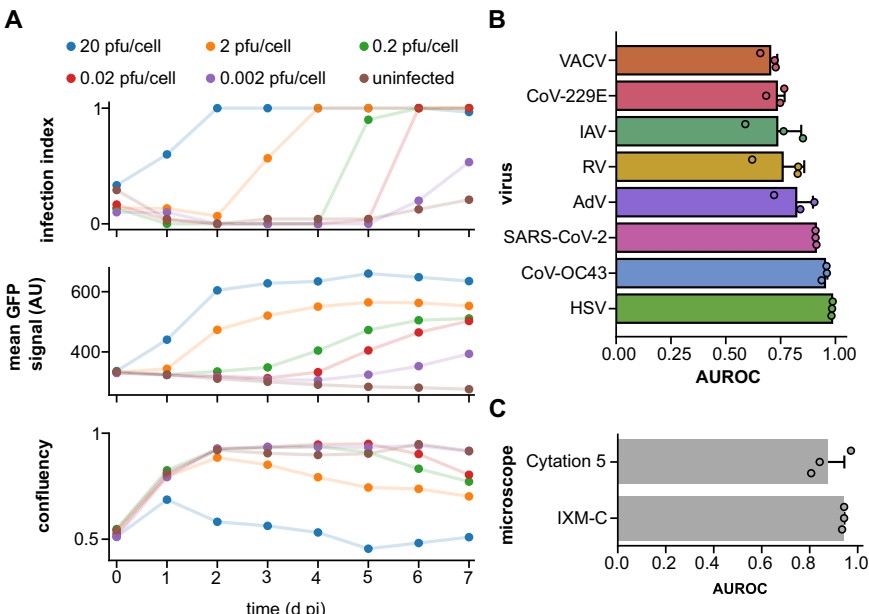

**Fig. 3 | Generalization of DVICE under new experimental settings. A** Time-resolved analysis of A549 cells infected with AdV-C5-IX-FS2A-GFP with quantification of predicted infection index by DVICE, GFP intensity, and cell confluency. pfu plaque forming units, AUROC area under the receiver operating characteristic curve. Data are presented as means. **B** Leave-one-out cross-validation of DVICE. Images of wells inoculated with the indicated virus were left out during network training. The performance was evaluated on the left-out images. Data are presented as means and error bars indicate standard deviations. $n = 3$. **C** Comparison of DVICE' performance between two microscopes, including ImageXpress Micro Confocal (IXM-C, Molecular Devices) and Cytation 5 (Agilent). $n = 480$ (216 infected and 264 uninfected images). Data are presented as means and error bars indicate standard deviations. Source data are provided as a Source Data file.

DVICE achieved an accuracy of $0.799 \pm 0.025$, an F1 score (harmonic mean of sensitivity and specificity) of $0.802 \pm 0.026$, and a Matthews Correlation Coefficient of $0.757 \pm 0.029$. Evaluation of the confusion matrix indicated that DVICE successfully recognized the different classes, with a slight bias towards AdV (Fig. 5B). Overall, DVICE achieved high sensitivity for all viruses (Fig. 5C).

To explore which image properties could facilitate class-specific recognition, we generated a synthetic dataset to probe for the network output across a range of conditions (Supplementary Fig. 5). The combination of high confluency and spatial autocorrelation was associated with increased rates of detection of the classes uninfected, SARS-CoV-2, and VACV, while images with low confluency and spatial autocorrelation were associated with AdV and RV. This could reflect the biological phenotype of the viruses, as VACV and SARS-CoV-2 infections lead to the formation of syncytia resulting in clusters of cells. We conclude that virus-class recognition is well feasible, although at present less robust than classification of the infection state.

## Discussion

The course of the SARS-CoV-2 pandemic has dramatically indicated the global lack of rapidly available and reliable procedures to score the infectivity of viral pathogens in clinical samples in a standardized manner. Available PCR-based diagnostics is highly sensitive but may lead to false positive results[40,41], as it measures genome equivalents and falls short of providing reliable infectivity scores in both acute and persistent infection settings[5]. In the case of enteroviruses and SARS-CoV-2, these limitations have been exposed by anti-viral compounds that reduce the production of progeny without affecting replication[39,42]. Further examples include the HIV protease inhibitor Nelfinavir blocking human adenovirus cell egress, or the oxidation–reduction modulator Provay Blue broadly inhibiting the release of infectious coronavirus progeny from infected primary human airway cell cultures[27,36,43,44].

Here we introduce an accurate automated framework for the broad detection of viruses using light microscopy. The DVICE

framework holds a fully automated, label-free, robust procedure for quantification of virus-induced CPE. It allows live monitoring of virus infections alone or in combination with fluorescence microscopy. DVICE is highly versatile and readily adaptable to new experimental settings, including new pathogens, cell lines, and imaging conditions. The current work demonstrates that DVICE can efficiently discriminate between viral lesions in a confluent monolayer and subconfluent conditions upon sparse cell seeding. This reflects the notion that virus-induced cell lesions are uniquely suited to be recognized by our framework. Our work extends previous analyses of subnuclear patterns of infected cells and the shape of cell lesions to predict lytic, cell-free, or non-lytic, cell-based viral transmission[12,44–46]. DVICE has the power to determine the nature of the virus based on a particular CPE signature. We propose that PCR-based assays are complemented with cell culture-based automated scoring of infectious units. Absence of detectable infectivity may provide important clinical milestones in disease management and diagnosis, including continuation, interruption, or change of treatment. In this sense, DVICE may complement ML models to analyze diverse viral sequence data towards developing a range of new diagnostic tools[47]. Yet importantly, DVICE provides viral infectivity data and thereby enhances personalized anti-viral treatment options. If combined with anti-viral treatments, this will allow on-target therapies and may contribute to reducing the propagation of drug-resistant viruses appearing in nature as well as in virus- and host-directed anti-viral treatments[48–51].

In addition to classical infection assays, an infection readout by DVICE can facilitate the quantification of neutralizing antibody titers, for example, in human samples using microneutralization or plaque reduction neutralization tests. Neutralizing antibodies are an indicator of protective immunity and provide essential information for vaccine development and public health[52,53]. Our framework can be used to detect virus-induced CPE in populations of cultured cells based on transmitted light microscopy images. To score virus infection, the cells must be susceptible to the particular virus. For optimal throughput, the workflow benefits from an automated microscope. In our time-

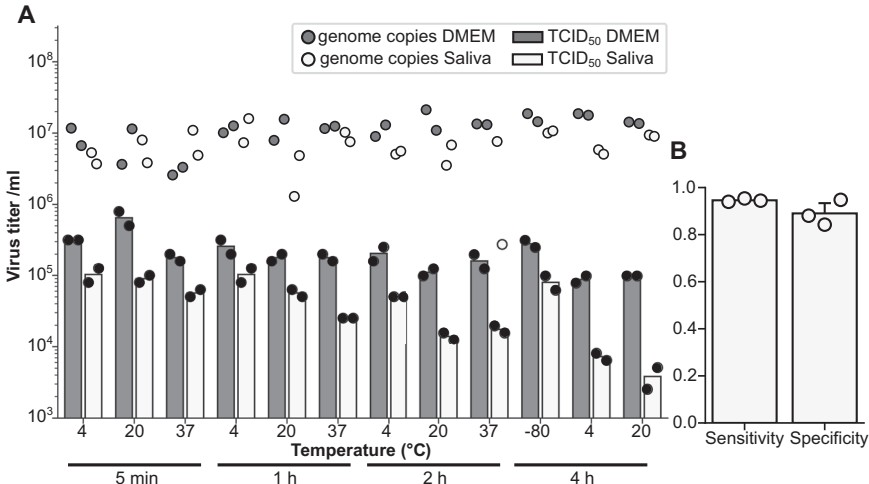

**Fig. 4 | SARS-CoV-2 infectious particle stability in saliva. A** Saliva and DMEM were spiked with SARS-CoV-2 and incubated at the indicated temperature and for the specified duration. The SARS-CoV-2 concentration after incubation was quantified by $TCID_{50}$ titration for infectious particles using the Reed–Muench method[7] (bars) and by RT-qPCR for virus genome copies (dots). $n = 2$. **B** Sensitivity and specificity values for DVICE classifications. Sensitivity (true positive rate) is defined as sensitivity $= \frac{TP}{TP+FN}$, and specificity (true negative rate) as specificity $= \frac{TN}{TN+FP}$, where TP = true positive, FN = false negative, TN = true negative, FP = false positive. Data include 3646 images, of which 627 were infected. Plot shows means ± standard deviations, $n = 3$. Source data are provided as a Source Data file.

resolved infection analysis, the onset of viral gene expression occurred before infection detection by DVICE, indicating that DVICE may be unable to detect the presence of a virus early in infection or a virus that persists in cells without overt cytopathic effects. This limitation is, however, compensated by the ultra-high sensitivity of DVICE, which allows for the detection of a single plaque-forming unit per well of a 96-well dish containing about $2 \times 10^4$ cells, equivalent to an MOI of <0.0001. We acknowledge that DVICE may not reach the same accuracy under experimental conditions different from the training conditions. Experimental variations may, for example, include different label-free imaging modalities, such as differential interference contrast microscopy, different magnifications, different cell lines, viral strains, or culture conditions. Nonetheless, and despite the widely recognized difficulty to generalize DL-based procedures, our framework is versatile and adaptable to unequivocally score different viruses in cell lines and imaging microscopes. This owes to robust learning of features associated with CPE, as shown by class activation mapping. To further enhance the adaptability of DVICE, we provide the user with resources for fine-tuning and transfer learning, broadly known procedures to increase AI accuracy with minimal additional input data[54].

## Methods
### Cell culture
Cell lines were cultivated in a T75 flask in Dulbecco's modified Eagle medium (DMEM, D6429; Sigma-Aldrich, St. Louis, USA) supplemented with 10% fetal bovine serum (FBS, 10270-106; Gibco, Carlsbad, USA) and non-essential amino acids (M7145; Sigma-Aldrich, St. Louis, USA). Cells were incubated in an environment of 37 °C, 5% $CO_2$, and 95% humidity. Cultures of VeroE6-TMPRSS2 were supplemented with 1 mg/mL geneticin (G418-RO, Merck), and cultures of A549-ACE2 and Huh7-ACE2 with 10 µg/mL blasticidin (ant-bl-1, InvivoGen) to preserve the expression of the transgenes. All cultures were passaged twice per week by washing with PBS and trypsinization (C-41020; Trypsin-EDTA, Sigma-Aldrich, St. Louis, USA).

### Viruses
AdV-C5 and AdV-C5-IX-FS2A-GFP were kindly provided by Silvio Hemmi (University of Zurich, Switzerland). CoV-229E-GFP, CoV-OC43, and SARS-CoV-2 were obtained as described previously[27].

Recombinant HSV-1-C12-CMV-GFP[55,56] was kindly provided by Stacey Efstathiou (University of Cambridge, UK). VACV_WR E/L-GFP was kindly provided by Jason Mercer (University of Birmingham, UK).

### Transmitted light and fluorescence live cell microscopy
Transmitted light images were acquired at 7 dpi using the high-throughput microscope ImageXpress Micro Confocal (IXM-C, Molecular Devices) with a ×4 air objective. Images had a resolution of 2048 × 2048 pixels and a depth of 16 bit. Fluorescence microscopy images were likewise acquired at the IXM-C. Images acquired at the Cytation 5 (Agilent) had a resolution of 1992 × 1992 pixels and a depth of 16 bit. Cells were imaged in a BSL-2 environment without fixation. Cells infected with SARS-CoV-2 were fixed by addition of paraformaldehyde to a final concentration of 4%. Plates were decontaminated and transferred to a BSL-2 laboratory for image acquisition.

### Infection assay and readout by crystal violet staining
For infection experiments, T75 flasks of 90% confluent cell cultures were trypsinated, and cells were re-suspended in 10 ml DMEM. Cells were diluted to a concentration of 100,000 cells per ml. 10,000 cells were seeded in 100 µl medium overnight. For sparsely seeded wells, 1000 cells were seeded in 100 µl, and images were acquired the next day.

Infection experiments were performed in a biosafety level (BSL)-2 laboratory, except for experiments with SARS-CoV-2, which were performed in a BSL-3 laboratory. 8 different virus concentrations were prepared by serial dilutions, and cells were infected by addition of 20 µl inoculum with 10 replicates per condition per plate. The first and last columns of each plate were left uninfected and supplemented by 20 µl fresh medium.

After image acquisition, cells were fixed by adding 30 µl of a 16% paraformaldehyde (PFA) solution for 30 min, except for SARS-CoV-2, where fixation was performed prior to acquisition. PFA was then discarded, and 50 µl of a 0.25% crystal violet (CV) staining solution prepared in an aqueous solution with 10% methanol was added for one hour. The CV staining solution was discarded, and cells were washed by submerging the plate in water, after which the plates were left to dry. The infection phenotype was independently assessed by three human experts, and image annotations were obtained by majority vote.

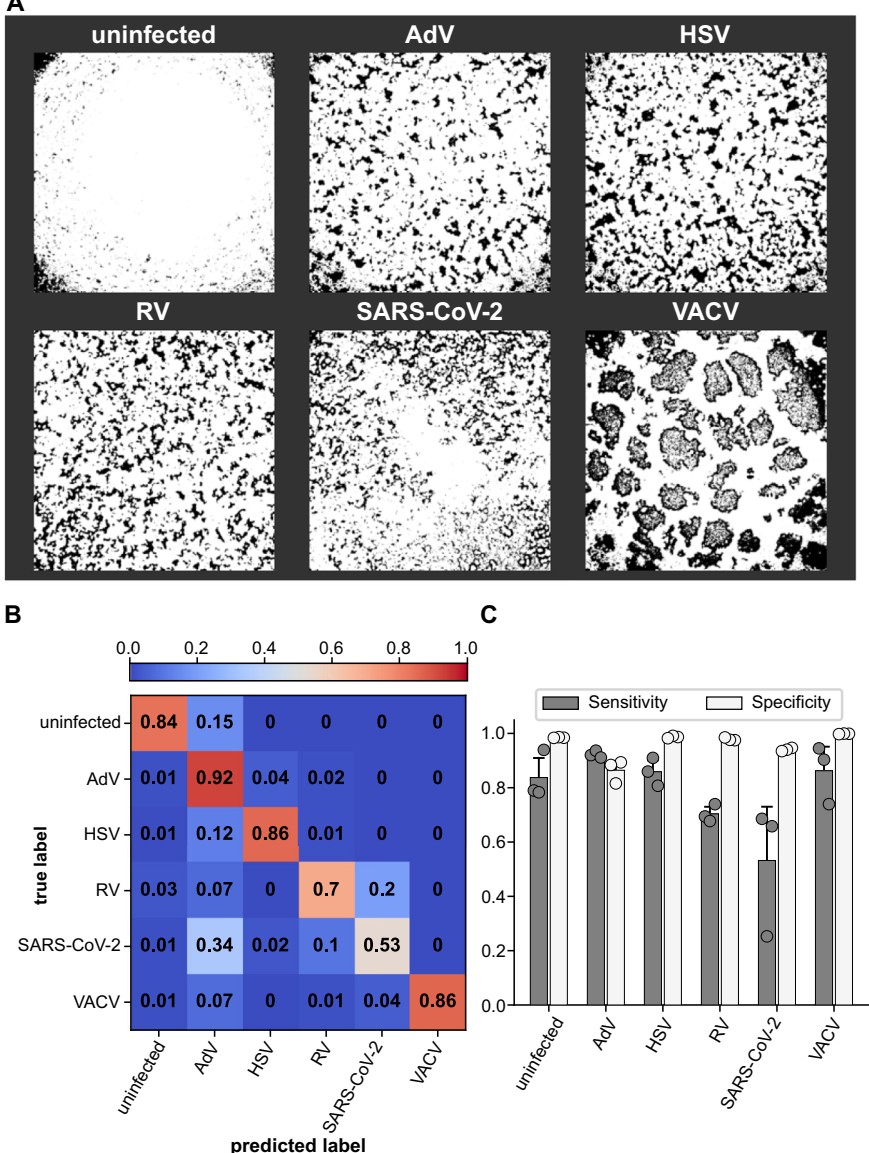

**Fig. 5 | Virus class-specific identification using DVICE. A** Example images of the segmentation maps used for virus class-specific identification. **B** Confusion matrix, indicating fractions of correct (across the diagonal) and incorrect classifications. Values are normalized to add up to 1 in each row except for cases of rounding errors. **C** Sensitivity (true positive rate) and specificity (true negative rate) of virus class detection by DVICE. Sensitivity was calculated as sensitivity $= \frac{TP}{TP+FN}$ and specificity as specificity $= \frac{TN}{TN+FP}$. TP = true positive, FN = false negative, TN = true negative, FP = false positive. Data are presented as means and error bars indicate standard deviations. $n = 3$. Source data are provided as a Source Data file.

## SARS-CoV-2 infectious particle stability in saliva

Saliva samples were collected from healthy, adult individuals who provided written informed consent. Saliva specimens (250 µl) were spiked with 50 µl of SARS-CoV-2 BA.1 variant. Spiked salivas were then diluted in DMEM medium at a final volume of 1 ml and filtered with a 0.22 µm Millex-GV Filter (Mercks) to eliminate bacteria. As a control, 50 µl of SARS-CoV-2 BA.1 were spiked in 950 µl DMEM medium and filtered similarly. SARS-CoV-2 spiked suspensions were then incubated at −80, 20, and 4 °C for 24 h; at 37, 20, and 4 °C for 2, 1, or 5 min. Each time point and temperature was tested with two independent biological replicates of spiked saliva and DMEM control. At the end of the incubation time, the respective virus suspensions were serially diluted in tenfold steps and inoculated on VeroE6-TMPRSS2 cells for virus $TCID_{50}$ titration. In parallel, 50 µl of the respective incubated virus suspensions were mixed with 150 µl of TRIzol reagent (Thermo Fisher) and subjected to RNA extraction with Direct-zol RNA Miniprep kit (Zymo Research) according to the manufacturer's protocol. Extracted SARS-CoV-2 RNAs were then quantified by RT-qPCR according to a previously described method[27,43]. The procedures involving human saliva did not fall under the Human Research Act according to the Ethical Board of the Kanton Zurich, Switzerland (BASEC number Req-2022-01020) and, therefore, did not require specific permission by a particular ethical board.

## Densitometric analysis of crystal violet staining

To quantify the absorbance of 96-well plates stained with crystal violet, a spectral scan of fully confluent wells was performed at a Tecan plate reader. Subsequent measurements of absorbance were performed at the spectral range with the highest absorbance, which was at 555 ± 4.5 nm.

## Dataset preparation and deep learning

Images were rescaled to an 8-bit range using min-max normalization, resized to 224 × 224 pixels using bicubic interpolation, and converted

to RGB format. Images were split into training, validation, and test sets in a stratified way while ensuring that the images reserved for the test set came from plates not present in the training or validation set. For infection, phenotype classification with DVICE, the tensorflow[57] (version 2.7) implementation of the EfficientNet-B0 architecture[26] was used. The model contains 7 distinct mobile inverted bottleneck convolution (MBConv)[58] blocks with squeeze-and-excitation (SE)[59] layers as attention mechanism. A custom head was added that comprises a global average pooling (GAP) layer, a dropout layer with a rate of 0.3, and a two-way dense layer for the final classification or a 6-way dense layer for virus class-specific classification. The network has 4 million parameters, which were randomly initialized. The comparatively low number of trainable parameters facilitates training and finetuning. Training was performed with a batch size of 128 on an NVIDIA GeForce RTX 3090. An Adam optimizer was used to minimize a class-weighted binary (or categorical in the case of virus class-specific classification) cross-entropy loss function with an initial learning rate of 0.001 for the first 10 epochs, after which the learning rate decayed by a factor of $e^{-0.1}$ every epoch. Training images were augmented by horizontal and vertical reflections. For the evaluation, the model from the epoch with the lowest loss on the validation set was selected.

### Training of machine learning models

For training of additional machine learning (ML) algorithms, histograms of oriented gradients (HOG)[29] were computed using scikit-image[60], resulting in 2592 features per image. HOG features were standardized, scaled to unit variance, and split to training and test data in a stratified way with threefold cross-validation. 90% of the data were selected for training and 10% for test purposes. ML algorithms included Gaussian naive Bayes (GNB), logistic regression (LR), $k$-nearest neighbor ($k$-NN), random forest (RF), decision tree (DT) classifier, and support vector machine (SVM) in their scikit-learn[61] implementation. Optimal parameters were determined by an initial grid search. SVM did not converge and was stopped after 1000 epochs when the present state of the model was used for evaluation. Statistical evaluation was performed using the nonparametric Kruskal–Wallis test with Dunn's correction for multiple tests in GraphPad PRISM (version 9.3.1).

### Model evaluation

Model performance was evaluated using the area under the receiver operating characteristic curve (AUROC). Ensembles of models were trained from different random seeds, which affected data selection and parameter initialization.

### Binary segmentation and cell density quantification

For cell density quantification, images were initially converted to 8-bit PNG images using min-max-normalization and rescaled to $1024 \times 1024$ pixels using bicubic interpolation. Pixel classification was performed using a decision tree model trained in ilastik[62] to perform semantic segmentation. Confluency was calculated by dividing the number of foreground pixels by the total pixel number. For training of the virus class-specific DVICE models, the segmentation maps were downscaled to $224 \times 224$ pixel images using bicubic interpolation.

### TCID$_{50}$ and plaque forming unit value calculation

TCID$_{50}$ and plaque forming unit (pfu) values were calculated using the specific infection (SIN) method, which provides a probabilistic estimate of a sample's infectivity[30]. The relationship between SIN or pfu values and TCID$_{50}$ values is provided by the Poisson distribution and was calculated as $1\,\mathrm{SIN} = 1\,\mathrm{pfu} = \frac{1}{\ln(2)}\,\mathrm{TCID}_{50} \approx 1.44\,\mathrm{TCID}_{50}$, where ln is the natural logarithm. The theoretical infection index is also provided by the Poisson distribution and was calculated as the probability $P$ of a well receiving at least one infectious particle $n$ and is provided by $P(n \geq 1) = 1 - P(n=0) = 1 - \exp(-\mathrm{pfu})$, where exp is the natural

exponential function. For Fig. 4, TCID$_{50}$ values were calculated using the Reed–Muench method[7].

### Resource availability

Further information and requests for resources and reagents should be directed to and will be fulfilled by the lead contacts, Prof. Dr. Urs Greber (urs.greber@mls.uzh.ch) and Dr. Anthony Petkidis (anthony.petkidis@uzh.ch).

### Reporting summary

Further information on research design is available in the Nature Portfolio Reporting Summary linked to this article.

## Data availability

The imaging data are available under restricted access due to legal considerations involving a patent application by the University of Zurich. Access can be obtained for non-commercial research and validation purposes upon agreement under an MTA by contacting the corresponding authors, U.F.G. (urs.greber@mls.uzh.ch) and A.P. (anthony.petkidis@uzh.ch) and will be provided within four weeks for academic use and restricted to the particular institution that requested access. Source data are provided with this paper.

## Code availability

The code used in this study is deposited at Zenodo at https://doi.org/10.5281/zenodo.11059621[63] and is available under restricted access due to legal considerations involving a patent application by the University of Zurich. Access can be obtained for non-commercial research and validation purposes upon agreement under an MTA by contacting the corresponding authors, U.F.G. (urs.greber@mls.uzh.ch) and A.P. (anthony.petkidis@uzh.ch) and will be provided within 4 weeks for academic use and restricted to the particular institution that 400 requested access.

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

## Acknowledgements

We acknowledge Prof. Ivo Sbalzarini, Dr. Artur Yakimovich and members of the Greber lab for helpful discussions, and Nicole Meili, Lucy Fischer, and Leta Fuchs for technical support with cell culture. The project was supported by the Swiss National Science Foundation Coronavirus Special Call to UFG (31CA30_196177/1) and the Pandemic Fund of the University of Zürich to UFG.

## Author contributions

A.P., V.A., R.V., and U.F.G. conceived and designed the experiments; A.P. and R.V. performed the experiments; A.P., L.M., and R.V. curated the data, A.P., V.A., L.M., R.V., and U.F.G. analyzed the data and contributed materials/analysis tools; A.P. and V.A. visualized the data, A.P. and U.F.G. wrote the paper. All authors discussed the results and commented on the manuscript.

## Competing interests

A.P., V.A., L.M., R.V., and U.F.G. filed a patent application, EP24168806.8, with the University of Zurich, entitled 'Method for labeling an image of a plurality of cells as having or not having a virus-induced cytopathic effect'. The patent application includes training of computational models derived from ensembles of cells infected with different viruses and imaged under described modalities, involving equipment and laboratory settings, such that the models recognize virus type-specific infection features. Patent pending.
