## [Peer Review File · Nature Communications]

REVIEWER COMMENTS

Reviewer #1 (Remarks to the Author):

The study presents DVICE, an AI framework for detection of virus-induced cytopathic effect (CPE). It is claimed that DVICE demonstrates virus class specificity and the author suggest that the framework can be adapted to laboratory diagnostics. DVICE represents an interesting and potentially promising machine learning diagnostic tool to be used complementary to the PCR-based assays in the clinics. However, I have a few concerns to be addressed by the authors.

1) My major concern is whether DVICE complex model really outperforms simpler models as it is suggested in Figure 2A. First, the errors bars seem to be nearly overlapping for DVICE and RF. Second, I would like to see more details on how the hyperparameters of the RF model were optimized, and what was the exact input for the RF model?

2) Could the authors consider another simple model which is based on comparing systematic differences in pixel intensities between the images for infected and uninfected cells? From the sample images in Figure 2D I got an impression that the uninfected images look quite different compared to the infected images. If this is a generalizable for other images (btw could the authors include more sample images?), perhaps a very simple model that just compares "average" pixel intensities between infected and uninfected images would also reach high AUROC?

3) Figure 2A: could the authors plot the actual ROC-curves instead of the AUROC stats for each machine learning algorithm?

4) Figure 2D, CAM images: from my experience, a CNN might often learn to differentiate between classes by the amount of background. Can the authors quantify or provide any other arguments on how much the amount of background present on the images affects the CNN classification?

5) Figure 5 A and B: could the authors provide confidence intervals for the computed sensitivity and specificity? How can you explain the low specificity of HSV?

6) Figure 5C: it is not clear to me what was the difference models 1-3 and why they demonstrate very different performance, could the authors provide more details and explanations?

Reviewer #2 (Remarks to the Author):

The authors present a deep neural network to detect cytopathic effects of different types of virus on cell cultures in 96 well plates using the convolutional neural network EfficientNet-B0. The authors compare the detection of virally induced of the trained neural network to classic machine learning approaches. The authors further describe the time course of detection of different viral dilutions over time after infection. Lastly the authors describe a variation of the approach, where the neural network is trained to distinguish between different virus infections.

Overall, it is an important topic to apply deep neural networks for new tasks to increase the throughput of biological data processing. Important contributions from the manuscript are that there are multiple virus-types and larger training datasets. Positive is that the authors are providing Google Colab files and will also make the data publicly available, which will help other labs to benefit from the code as well as the raw data.

There are several concerns about the manuscript in its current form:

1) I assume there has been a decision at some point to choose one deep neural network architecture, which would be important to add to the manuscript why this architecture was chosen.

2) AUROC is not an ideal metric to evaluate the model performance. It would be important to use metrics that also include True negatives and False negatives?

3) Figure 5 C: I was trying to find out what the 3 models in this Figure refer to. Can the authors explain if these are 3 different models that were trained with the same dataset?

If this is the case, what are the faded lines showing?

And it looks like model 1 is not converging. Could the authors show, if this is happening more often with the EfficientNet architecture?

It would be very important to see training and evaluation graphs for the other tasks, to understand the performance better. (see point 1, about if this is the right architecture)

4) In the abstract the authors write: 'Strikingly, DVICE exhibits virus class specificity, as shown with adenovirus, herpesvirus and influenza virus in human lung epithelial cells.' Whereas later when presenting the data, the authors state:

'Evaluation of specificity and recall showed that DVICE can learn virus-specific signatures considerably better than chance, albeit with deviations between different viruses and models (Figure 5A, 5B). Notably, a trend to the latter was already observed during network training (Figure 5C). We conclude that virus-class recognition is well feasible, although at present less robust than classification of the infection state.'

The discussion of the data is definitely a more realistic assessment of the data. However, it is not state-of-the-art that class recognition is above chance.

It is of course a much more difficult task to classify which virus, which might not be task that is routinely performed. I would recommend the authors adjust their statement in the abstract to reflect, that it is to some degree possible to indicate which virus infected the cells, even though the reliability is low.

Again here, it would be useful to identify and show the type of errors occurring. It would be particularly interesting if certain misclassifications occur more frequently than others, meaning if certain virus classes are often confused by the network.

5) In Figure 4B HSV shows the highest AUROC whereas in in Figure 5 HSV has extremely low specificity and sensitivity. Could the authors comment where the discrepancy is coming from?

REVIEWER COMMENTS

Reviewer #1 (Remarks to the Author):

The study presents DVICE, an AI framework for detection of virus-induced cytopathic effect (CPE). It is claimed that DVICE demonstrates virus class specificity and the authors suggest that the framework can be adapted to laboratory diagnostics. DVICE represents an interesting and potentially promising machine learning diagnostic tool to be used complementary to the PCR-based assays in the clinics. However, I have a few concerns to be addressed by the authors.

1) My major concern is whether DVICE complex model really outperforms simpler models as it is suggested in Figure 2A. First, the errors bars seem to be nearly overlapping for DVICE and RF. Second, I would like to see more details on how the hyperparameters of the RF model were optimized, and what was the exact input for the RF model?

We thank the reviewer for the comment and have extended our comparison between DVICE and conventional machine learning methods. We now provide two new supplementary figures exploring the feasibility of confluency- and intensity-based image classification (please also see reply to the next comment).

The data preparation for the machine learning models is described in the methods section. Briefly, histograms of oriented gradients (HOG) were computed using scikit-image, resulting in 2592 features per image. HOG features were standardized, scaled to unit variance, and split to training and test data in a stratified way with threefold cross validation. Ninety percent of the data were selected for training and 10 % for test purposes. We found this approach to be superior to intensity- or confluency-based image classification.

We are now including a new supplementary table that specifies the grid used in the search. During the revision period, we have optimized the hyperparameters of the conventional machine learning models, leading to improved performance and reproducibility. We have included a one-way analysis of variance (ANOVA) with multiple comparisons between DVICE and conventional machine learning models, indicating that DVICE still significantly outperforms the latter. We replaced panel A of Figure 2 as follows:

Figure 2: Network performance and characteristics. (A) Area under the receiver operating characteristic curve (AUROC) for different machine learning algorithms, including support-vector machine (SVM), *k*-nearest neighbors (*k*-NN), Gaussian naive Bayes (GNB), decision tree (DT), logistic regression (LR), random forest (RF), and DVICE. A one-way analysis of variance (ANOVA) with multiple comparisons was performed to evaluate the conventional ML algorithms against DVICE. Adjusted p-values: **** = $p < 0.0001$.

Supplementary table 1: Parameters of machine learning models used in grid search

Machine learning model	Parameter	Value range
Gaussian Naïve Bayes (GNB)	var_smoothing	numpy.logspace(0, -9, num = 100), equivalent to $\{10^{-9i} i = 0, 1, 2, \dots, 99\}$
Random Forrest Classifier (RF)		
Logistic Regression (LR)	solver	["lbfgs", "saga"]
	C	[1, 10]
Decision Tree Classifier (DT)	criterion	["gini", "entropy"]
	splitter	["best", "random"]
	max_depth	[5, 15, None]
Support Vector Machine (SVM)	kernel	["linear", "rbf", "poly"]
	C	[1, 10]
k nearest neighbor (k -NN)	n_neighbors	[3, 5, 10]

Additionally, we would like to point out that a random forest classifier also finds the following use in our manuscript: We have previously trained a model in ilastik for semantic image segmentation by providing ground truth annotations for foreground/background regions. In the new supplementary figure 3, we are now expanding on the utility of cell confluency as a feature for infection classification (please see Supplementary Figure 3 of the revised manuscript).

2) Could the authors consider another simple model which is based on comparing systematic differences in pixel intensities between the images for infected and uninfected cells? From the sample images in Figure 2D I got an impression that the uninfected images look quite different compared to the infected images. If this is generalizable for other images (btw could the authors include more sample images?), perhaps a very simple model that just compares "average" pixel intensities between infected and uninfected images would also reach high AUROC?

We have extended Supplementary Figure 1 to include three images per virus and uninfected cell line. We also address the question if a classifier based on image intensity can achieve similar performance in the new Supplementary Figure 4. In addition to average pixel intensity, we also employed the standard deviation in pixel intensity as another feature. We used the same grid search as for the HOG-based image classification to optimize the hyperparameters of the ML models. The best-performing model was *k*-nearest neighbors, which achieved an AUROC of around 0.8.

We have additionally randomly permuted the pixels in the input image and trained our DVICE model on this input. Here, DVICE still had access to the full range of pixel values but lost all spatial information. This resulted in a performance comparable to the other machine learning models, but was inferior to the performance achieved on normal images.

Supplementary Figure 4: Intensity-based image classification. (A) Mean intensity of images used in the study. (B) Standard deviations of image intensities of images used in the study. (C) Standardized mean intensity and standard deviation of intensities. Each dot represents one image. (D) Data were split into 90 % training data and 10 % test data in a random, stratified way and machine learning models were trained to classify images into infected or uninfected based on the image intensity and standard deviation provided in C. Data for model training was prepared using a fivefold, stratified shuffle split and optimal model parameters were determined using a grid search. SVM = Support Vector Machine, LR = Logistic Regression classifier, GNB = Gaussian Naïve Bayes, RF = Random Forest classifier, DT = decision tree classifier, k-NN = k-nearest neighbors. AUROC = area under the receiver operating characteristic curve. (E) Receiver operating characteristic (ROC) curves for the three best performing models. The dotted line indicates a random classifier. (F) DVICE models were trained on either normal (n) or randomly permuted (p) images. Model evaluation was performed on a withheld test set which also consisted of normal or permuted images.

3) Figure 2A: could the authors plot the actual ROC-curves instead of the AUROC stats for each machine learning algorithm?

We have now included plots of the ROC curves as part of Supplementary Figure 2.

Supplementary Figure 2A: Performance characteristics of ML models. (A) Receiver operating characteristic (ROC) curves of ML models from Figure 2A. SVM = Support Vector Machine, LR = Logistic Regression classifier, GNB = Gaussian Naïve Bayes, RF = Random Forest classifier, DT = decision tree classifier, k-NN = k-nearest neighbors. The dotted line indicates a random classifier.

4) Figure 2D, CAM images: from my experience, a CNN might often learn to differentiate between classes by the amount of background. Can the authors quantify or provide any other arguments on how much the amount of background present on the images affects the CNN classification?

We thank the reviewer for this interesting consideration. We are now addressing this question as part of Supplementary Figure 4F (please see above), which shows that random permutation of pixels in the image severely compromises DVICE' ability to learn correct infection classification. Thus, DVICE does not only leverage pixel intensity/number of background pixels present in the image, but also harnesses their contextual location.

During our research we have also performed occlusion tests, for which an example is provided below. In images with spatially highly confined cytopathic effect (CPE), we observed that DVICE specifically focused on these regions. We then covered the CPE-associated region either by a black or white box, or by mean inpainting. As a result, we observed that occluding the CPE-associated image region abrogated infection detection and led to a diffuse CAM. We have decided not to include these data in our manuscript, as the number of example images with highly confined lesions in our dataset was very limited, and occlusion with a black box arguably creates an image outside of the sampling distribution.

input image

CAM
original

overlay

occluded

5) Figure 5A and B: could the authors provide confidence intervals for the computed sensitivity and specificity? How can you explain the low specificity of HSV?

Thank you for the comment. We have modified our approach to virus class-specific classification and have therefore adjusted Figure 5. The new version of the figure also includes standard deviations for the computed sensitivity and specificity (corresponding to a 68 % confidence interval).

Figure 5: Virus class-specific classification using DVICE. (A) Example images of the segmentation maps used for virus class-specific classification. (B) Confusion matrix, indicating fractions of correct (across the diagonal) and incorrect classifications. Values are normalized to add up to 1 in each row except for cases of rounding errors. (C) Sensitivity (true positive rate) and specificity (true negative rate) of virus class detection by DVICE. Sensitivity was

calculated as $TP/(TP+FN)$ and specificity as $TN/(TN+FP)$ TP = true positive, FN = false negative, TN = true negative, FP = false positive. n = 3

6) Figure 5C: it is not clear to me what was the difference models 1-3 and why they demonstrate very different performance, could the authors provide more details and explanations?

Thank you for the comment. The original models varied in the assignment of data to the training and validation sets, but during the revisions we discovered a problem with data stratification in one of the models, which resulted in poor performance. We have modified our approach to virus class-specific classification, as discussed in the revised manuscript.

Reviewer #2 (Remarks to the Author):

The authors present a deep neural network to detect cytopathic effects of different types of virus on cell cultures in 96 well plates using the convolutional neural network EfficientNet-B0. The authors compare the detection of virally induced of the trained neural network to classic machine learning approaches. The authors further describe the time course of detection of different viral dilutions over time after infection. Lastly the authors describe a variation of the approach, where the neural network is trained to distinguish between different virus infections. Overall, it is an important topic to apply deep neural networks for new tasks to increase the throughput of biological data processing. Important contributions from the manuscript are that there are multiple virus-types and larger training datasets. Positive is that the authors are providing Google Colab files and will also make the data publicly available, which will help other labs to benefit from the code as well as the raw data. There are several concerns about the manuscript in its current form:

1) I assume there has been a decision at some point to choose one deep neural network architecture, which would be important to add to the manuscript why this architecture was chosen.

We thank the reviewer for this suggestion. Indeed, we have initially experimented with a variety of CNNs, including ResNets and MobileNets. We settled on EfficientNets, as this architecture contains comparatively few trainable parameters (around 4 million), which allows fast training and finetuning. We have now added a corresponding explanation to the text.

2) AUROC is not an ideal metric to evaluate the model performance. It would be important to use metrics that also include True negatives and False negatives?

Thank you for the comment. In the new supplementary table 2 (please see below), we are now including other metrics to characterize model performance and provide the ROC curves in Supplementary Figure 2A (please see above), which display the True Positive Rate and False Positive Rate.

Supplementary table 2: Performance metrics of machine learning models. Accuracy indicates the fraction of correctly classified images, given by $Accuracy = \frac{TP+TN}{TP+TN+FP+FN}$. The F1 score, also known as Sørensen–Dice coefficient, is the harmonic mean of precision and recall, given by $F1 = \frac{2 \cdot precision \cdot recall}{precision+recall}$, where $precision = \frac{TP}{TP+FP}$ and $recall = \frac{TP}{TP+FN}$. The Matthews correlation coefficient (MCC) reflects the correlation between observed and predicted classes and is given by $MCC = \frac{TP \cdot TN - FP \cdot FN}{\sqrt{(TP+FP)(TP+FN)(TN+FP)(TN+FN)}}$. The area under the receiver operator characteristic (ROC) curve (AUROC) is the integral of the function. TP = true positive, TN = true negative, FP = false positive, FN = false negative. Data indicate means \pm standard deviation, n = 3

metric	SVM	k-NN	DT	GNB	LR	RF	DVICE
Accuracy	0.4246 \pm 0.0044	0.6301 \pm 0.0033	0.7473 \pm 0.0143	0.7788 \pm 0.0008	0.8848 \pm 0.0018	0.8803 \pm 0.0003	0.9912 \pm 0.0012

F1 score	0.4279 ± 0.0073	0.5383 ± 0.0048	0.7386 ± 0.0126	0.7662 ± 0.0008	0.8837 ± 0.0016	0.8736 ± 0.0002	0.9912 ± 0.0012
AUROC	0.4353 ± 0.0056	0.5844 ± 0.0042	0.7879 ± 0.0232	0.8489 ± 0.0016	0.9404 ± 0.0027	0.9429 ± 0.0011	0.9914 ± 0.0012
MCC	-0.1190 ± 0.0077	0.1286 ± 0.0133	0.4522 ± 0.0307	0.5259 ± 0.0022	0.7550 ± 0.0039	0.7455 ± 0.0008	0.9825 ± 0.0024

3) Figure 5C: I was trying to find out what the 3 models in this Figure refer to. Can the authors explain if these are 3 different models that were trained with the same dataset? If this is the case, what are the faded lines showing? And it looks like model 1 is not converging. Could the authors show, if this is happening more often with the EfficientNet architecture? It would be very important to see training and evaluation graphs for the other tasks, to understand the performance better. (see point 1, about if this is the right architecture)

Thank you for asking to clarify this point. We have now added plots for the training and validation loss and accuracy for the binary DVICE model in Supplementary Figure 2B

Supplementary Figure 2B: Training behavior of DVICE. Graphs show binary cross entropy loss and accuracy of DVICE model on training and validation dataset. Shaded regions indicate standard deviation, $n = 3$.

4) In the abstract the authors write: 'Strikingly, DVICE exhibits virus class specificity, as shown with adenovirus, herpesvirus and influenza virus in human lung epithelial cells.' Whereas later when presenting the data, the authors state: 'Evaluation of specificity and recall showed that DVICE can learn virus-specific signatures considerably better than chance, albeit with deviations between different viruses and models (Figure 5A, 5B). Notably, a trend to the latter

was already observed during network training (Figure 5C). We conclude that virus-class recognition is well feasible, although at present less robust than classification of the infection state.' The discussion of the data is definitely a more realistic assessment of the data. However, it is not state-of-the-art that class recognition is above chance. It is of course a much more difficult task to classify which virus, which might not be task that is routinely performed. I would recommend the authors adjust their statement in the abstract to reflect, that it is to some degree possible to indicate which virus infected the cells, even though the reliability is low.

We thank the reviewer for the thoughtful comment, and have adjusted the statement accordingly.

Again here, it would be useful to identify and show the type of errors occurring. It would be particularly interesting if certain misclassifications occur more frequently than others, meaning if certain virus classes are often confused by the network.

This is an interesting point, which we now address as part of Figure 5 in a confusion matrix. As anticipated by the reviewer, our networks have a slight bias towards the class AdV. Curiously, this was even the case as this virus class was underrepresented in the training dataset compared to the other viruses.

5) In Figure 4B HSV shows the highest AUROC whereas in Figure 5 HSV has extremely low specificity and sensitivity. Could the authors comment where the discrepancy is coming from?

Figure 3B shows a leave-one-out cross validation, where we investigated the feasibility for DVICE to discover "new" virus infection phenotypes. High performance for HSV in the left-out virus experiment would indicate that the HSV infection phenotype is similar to the one of the remaining viruses. This suggests that the HSV infection phenotype is not particularly uncommon, a feature which may make it harder to recognize in virus class-specific classification. In the revised version of the manuscript, we have modified our approach to virus class-specific classification. Here, HSV shows a similar performance as AdV and VACV other viruses.

REVIEWERS' COMMENTS

Reviewer #1 (Remarks to the Author):

Thank you for thoroughly addressing my questions, well done, I have no more questions.

Reviewer #1 (Remarks on code availability):

Well done

Reviewer #2 (Remarks to the Author):

The authors have addressed my previous concerns in the revised manuscript.